Quorum sensing signals of the grapevine crown gall bacterium, Novosphingobium sp. Rr2-17: use of inducible expression and polymeric resin to sequester acyl-homoserine lactones

Gan Han Ming 1 2
http://orcid.org/0000-0003-3204-9565 Dailey Lucas 3
Wengert Peter 3
http://orcid.org/0000-0003-3100-5704 Halliday Nigel 4
http://orcid.org/0000-0002-1920-5036 Williams Paul 4
Hudson André O. 3
http://orcid.org/0000-0003-1328-2978 Savka Michael A. 3 massbi@rit.edu
1 Patriot Biotech Sdn Bhd , Subang Jaya, Selangor , Malaysia
2 Department of Biological Sciences, Sunway University , Bandar Sunway, Petaling Jaya , Malaysia
3 The Thomas H. Gosnell School of Life Sciences, Biotechnology and Molecular Bioscience Program, College of Science, Rochester Institute of Technology , Rochester, New York , United States
4 Biodiscovery Institute and School of Life Sciences, University of Nottingham , Nottingham , United Kingdom
Bolshoy Alexander
Electronic publication date: 2024 Dec 20
Publication date: 2024
Volume: 12
Electronic Location ID: e18657
Received 2024 Jun 17; Accepted 2024 Nov 17
Copyright: © 2024 Gan et al.
Copyright year: 2024
Copyright holder: Gan et al.
License: This is an open access article distributed under the terms of the Creative Commons Attribution License, which permits unrestricted use, distribution, reproduction and adaptation in any medium and for any purpose provided that it is properly attributed. For attribution, the original author(s), title, publication source (PeerJ) and either DOI or URL of the article must be cited.
License URL: https://creativecommons.org/licenses/by/4.0/

Keywords: Acyl-homoserine lactones, Novosphingobium. sp., Quorum sensing, Inducible expression, Resin, NovI, Grapevine crown gall tumor, Agrobacterium vitis tumor

Funding: American Society for Microbiology through an Undergraduate Research Fellowship Lucas Dailey. Lucas Dailey and Rochester Institute of Technology Paul Williams’ laboratory at the University of Nottingham, UK Monash University Malaysia Tropical Medicine and Biology Multidisciplinary Platform Funding was provided by the American Society for Microbiology through an Undergraduate Research Fellowship to Lucas Dailey. Lucas Dailey and Rochester Institute of Technology (RIT) provided the use of facilities and supplies to perform experiments. Paul Williams’ laboratory at the University of Nottingham, UK, provided the LC-MS/MS analysis and synthesis of AHL standards carried out by Alex Truman. Han Ming Gan received support from the Monash University Malaysia Tropical Medicine and Biology Multidisciplinary Platform. DNA primers, DNA isolation and DNA purification kits, enzymes and other expendables were provided by the Thomas H. Gosnell School of Life Sciences. The funders had no role in study design, data collection and analysis, decision to publish, or preparation of the manuscript.

==============================
Background

A grapevine crown gall tumor strain, Novosphingobium sp. strain Rr2-17 was previously reported to accumulate copious amounts of diverse quorum sensing signals during growth. Genome sequencing identified a single luxI homolog in strain Rr2-17, suggesting that it may encode for a AHL synthase with broad substrate range, pending functional validation. The exact identity of the complete suite of AHLs formed by novIspR1 is largely unknown.

Methods

This study validates the function of novIspR1 through inducible expression in Escherichia coli and in the wild-type parental strain Rr2-17. We further enhanced the capture of acyl homoserine lactone (AHL) signals produced by novIspR1 using polymeric resin XAD-16 and separated the AHLs by one- and two-dimensional thin layer chromatography followed by detection using AHL-dependent whole cell biosensor strains. Lastly, the complete number of AHLs produced by novIspR1 in our system was identified by LC-MS/MS analyses.

Results

The single LuxI homolog of N. sp. Rr2-17, NovIspR1, is able to produce up to eleven different AHL signals, including AHLs: C8-, C10-, C12-, C14-homoserine lactone (HSL) as well as AHLs with OH substitutions at the third carbon and includes 3-OH-C6-, 3-OH-C8-, 3-OH-C10-, 3-OH-C12- and 3-OH-C14-HSL. The most abundant AHL produced was identified as 3-OH-C8-HSL and isopropyl-D-1-thiogalactopyranoside (IPTG) induction of novIspR1 expression in wild type parental Rr2-17 strain increased its concentration by 6.8-fold when compared to the same strain with the vector only control plasmid. Similar increases were identified with the next two most abundant AHLs, 3-OH-C10- and unsubstituted C8-HSL. The presence of 2% w/v of XAD-16 resin in the growth culture bound 99.3 percent of the major AHL (3-OH-C8-HSL) produced by IPTG-induced overexpression of novIspR1 in Rr2-17 strain. This study significantly adds to our understanding of the AHL class of quorum sensing system in a grapevine crown gall tumor associated Novosphingobium sp. Rr2-17 strain. The identity of nine AHL signals produced by this bacterium will provide a framework to identify the specific function(s) of the AHL-mediated quorum-sensing associated genes in this bacterium.

Introduction

Many bacteria modify their physiology in a cell density-dependent manner by producing and sensing extracellular signaling molecules, a process known as quorum sensing (QS). One of the most common QS mechanisms involves acyl-homoserine lactone (AHL) signals, allowing bacteria to coordinate population-level gene regulation in response to AHL signals reaching a threshold concentration (Azimi et al., 2020; Mukherjee & Bassler, 2019; Fuqua & Greenberg, 2002). As the bacterial population reaches a certain cell density, the concentration of AHL signal accumulates, triggering a receptor protein to regulate gene expression. These AHL signals are synthesized by LuxI-type proteins and detected by LuxR family transcriptional regulators (Fuqua, Winans & Greenberg, 1994). A typical AHL-QS system consisting of an LuxI and LuxR, are usually located in close genomic promiximity. Upon activation of the LuxR receptor, various population-wide responses can be initiated. Examples of such responses include the induction of conjugation in the tumor-inducing plasmid involved in the agrobacteria plant crown gall tumor disease. In other genera, this system leads to the production of virulence factors, biofilms, bioluminescence, and antimicrobial metabolites (Fuqua & Greenberg, 2002; Miller & Bassler, 2001; Piper, Beck von Bodman & Farrand, 1993; Waters & Bassler, 2005).

Many bacteria contain additional luxR transcriptional regulators that are not paired with a canonical luxI/luxR pair. These unpaired luxR genes, known as orphans or solos, are orthologs of the QS LuxR-type transcriptional regulators. LuxR solos are characterized by at least a DNA-binding helix-turn-helix (HTH) domain at the C terminus (Fuqua, 2006; Subramoni & Venturi, 2009). Initially, the N-terminal domain of LuxR solos was shown to bind AHLs and thereby broadening regulatory range by eavesdropping on exogenous signals produced by other bacteria in the vicinity. Further, some LuxR solos respond to plant-derived compounds and are referred to as plant-associated bacteria (PAB) LuxR solos. More recently, it has been discovered that LuxR solos can also interact with bacterial- and plant-derived signaling molecules other than AHLs. These include include pyrones, dialkylresorcinols, and ethanolamine derivatives (Brameyer et al., 2015; Al-Khdhairawi et al., 2019; Coutinho et al., 2018).

Some members of the LuxI protein family catalyze synthesis of AHL quorum sensing signals from S-adenosyl-L-methionine (SAM) and an acyl thioester. LuxI family members prefer acyl-CoA, and others prefer acyl-ACP (acyl carrier protein (ACP)) as the acyl-thioester substrate (Montebello et al., 2014). Although SAM is a conserved substrate for AHL synthases, specificity in the AHL signal is determined by the structure of the acyl chain, which can vary in length (short-chain vs. long-chain) and substitution (substituted vs. unsubstituted) in the acyl-ACP substrate. To achieve tight AHL signal specificity, AHL synthases must be selectively recognize the correct acyl-ACP substrate from the cellular acyl-ACP pool to synthesize the native autoinducer. However, the molecular basis of substrate selectivity in AHL synthases remains to be elucidated (Churchill & Chen, 2011).

Previous work by our group has isolated, sequenced and annotated the genome of an AHL-producing Novosphingobium sp. Rr2-17 isolated from a grapevine tumor and we have previously confirmed the function of a predicted luxI homolog (named novIspR1) in the Novosphigobium sp. Rr2-17 strain (Gan et al., 2009, 2012). However, novIspR1 has not been experimentally validated. During our initial studies, N. sp Rr2-17 strain accumulated large amounts of AHL signals when compared to other sphingomonad strains tested in our laboratory, albeit containing only a single predicted luxI homolog, novIspR1, in its genome.

Here we show by: (1) inducible homologous overexpression of novIspR1 in the parental wild type strain, (2) use of a polymeric resin for AHL signal recovery, (3) 1-D and 2-D thin layer chromatography to visualize AHLs, and (4) LC-MS/MS that the single novIspR1 directs the synthesis of nine AHL signals, with OH-C8 AHL being the most abundant signal. The inducible homologous overexpression of AHL synthase in the parent strain coupled with using polymeric Amberlite XAD-16 resin to preferably adsorb the AHL signals and using 2-D TLC with bioluminescent reporters allows the detection of minor AHLs that might otherwise elude detection and structural identification using conventional methods.

Materials and Methods

Bacterial strains, plasmids, growth media and biosensor strains

Novosphingobium sp. Rr2-17 was isolated from a nopaline grapevine tumor (Fig. 1) by Ernő Szegedi previously of the Research Institute for Viticulture and Enology, Kecskemét Hungary. Table one contains the bacterial strains, plasmids and primers used in this work (Table 1). Novosphingobium sp. Rr2-17 strain was grown in tryptone soy broth (TSB), potato dextrose (PD) or R2A medium (Difco Laboratories, Detroit, MI, USA) at 28 °C. Agrobacteria AB minimal medium (Chilton et al., 1974) at 28 °C was used to grow AHL-dependent biosensor strains Agrobacterium NTL4 and A136. For AHL signal induction bioassays, parental wild-type strain N. sp. Rr2-17 harboring empty vector pSRKKm or pSRKKm with cloned novIspR1 gene were grown on Luria-Bertani broth (LB) medium at 28 °C and supplemented with kanamycin at 50 mg/ml, respectively. For AHL signal detection bioassays, Agrobacterium tumefaciens NTL4 (pZLR4) and A136 (pCF218, pMV26) were grown in AB medium supplemented with 0.2% (w/v) dextrose and 0.01% (w/v) yeast extract and gentamycin (10 μg/ml) for NTL4 (pZLR4) (Cha et al., 1998) and kanamycin (25 μg/ml) and tetracycline (5 μg/ml) for A136 (pCF218, pMV26) (Sokol et al., 2003; Zhu et al., 2003). Escherichia coli-based biosensors JM109 (pSB401), JM109 (pSB1075) and JM109 (pSB536) were grown in LB medium with the appropriate antibiotic for plasmid maintenance (Swift et al., 1997; Winson et al., 1998). Chromobacterium violaceum CV026 biosensor was grown in tryptone yeast extract/potato dextrose (1:1) agar medium for T-streak bioassays (McClean et al., 1997). Each AHL-dependent bacterial biosensor strain used in this work along with its AHL receptor protein, cognate AHL signal and reporter gene output is listed in Table S1. All media and growth conditions for AHL detection bioassays are as previously described by our laboratory (Scott et al., 2006; Gan et al., 2009, 2016; Lowe et al., 2009).

Figure 1 Crown gall tumors.

Table 1 Bacterial strains, plasmids and primers used in this study.

Strain	Description	Ref	
Novosphingobium sp. Rr 2-17	Crown gall isolate, AHL-producer	Gan et al. (2009, 2012), this study	
Escherichia coli JM109	(traD36, pro AB+ lac Iq, laczΔM15) end A1 hsdR17 (rk−, mk+) mcrA supE44 γ - gyrA96 relA1Δ(lac-proAB)	Yanisch-Perron, Vieira & Messing (1985)	
Agrobacterium tumefaciens NTL4	pTiC58-cured derivative of C58ΔtetRS containing pZLR4 (traR, PtraG::lacZ) cognate AHL: 3-oxo-C8-HSL	Shaw et al. (1997)	
Agrobacterium tumefaciens A136	Ti plasmidless host, containing pCF218 (traR) and pMV26 (PtraI::luxCDABE, cognate AHL: 3-oxo-C8-HSL	Chambers et al. (2005)	
Chromobacterium violaceum CV026	Indicator strain for detection of alkanoyl-AHLs, derivative of wild-type strain 31532 with mini-Tn5, Kmr, in cviI, cognate AHL: C6-HSL	McClean et al. (1997)	
Plasmid	Feature	Ref	
pSRKKm	KmR, IPTG-inducible	Khan et al. (2008)	
pSRKKm::NovINsp	KmR, IPTG-inducible containing novI	This study	
Primer (Target gene)	Sequence and binding site	Ref	
NovIF (NovINsp)	Contig97 (13,765–13,784 bp)
GGAATTCCATatgatccattgcctttccaa	This study	
NovIR (NovINsp)	Contig97 (14,428–14,446 bp)
CCTAGGCTAGCcattcatccccgttgcttc	This study	

Protein alignment and phylogenetic tree analysis

LuxI homologs from the genus Novosphingobium were downloaded from the uniport database (as of 30th October 2022). Alignment of the amino acid sequences was performed with MUSCLE v.3.8.1551 (default settings) followed by trimming with trimAl v1.4 (-automated1 option) that selects alignment sites based on similarity statistics optimized for maximum likelihood phylogenetic tree reconstruction (Capella-Gutiérrez, Silla-Martínez & Gabaldón, 2009; Edgar, 2004). Maximum likelihood tree was subsequently inferred using FastTree with the Le-Gascuel 2008 model (-lg option) (Price, Dehal & Arkin, 2010). Visualization of the amino acid alignment and phylogenetic tree used JalView (https://www.jalview.org/) and FigTree v1.4.4 (https://github.com/rambaut/figtree/), respectively.

Amplification and cloning of novINsp.Rr2-17

Amplification of the novIspR1 was performed using Q5 polymerase mastermix (New England Biolabs, Ipswich, MA, USA) according to the manufacturer’s instructions. Approximately 150 ng of the purified PCR amplicons were mixed with 50 ng of pSRKKm vector (Khan et al., 2008) and double digested with NheI and NdeI (New England Biolabs, Ipswich, MA, USA) for 1 h. After heat inactivation, the digested products were purified using magnetic beads (Omega Biotek, Norcross, GA, USA) and ligated with Electroligase (New England Biolabs, Ipswich, MA, USA) for 30 min. The ligated products were transformed into wildtype N. sp. Rr2-17 and Escherichia coli JM109 using electroporation all as previously described by our laboratory (Scott et al., 2006; Gan et al., 2016).

Inducible expression of novINsp.Rr2-17 and use of Amberlite XAD-16 resin for capture of AHLs

AHL extractions from induction assays Rr2-17 (pSRKKm:: novIspR1) and Rr2-17 (pSRKKm) were grown in 20 mL of LB (50 mg/ml kanamycin) supplemented with different amount of isopropyl-D-1-thiogalactopyranoside (IPTG) inducer to final concentration of 0, 10, 100, or 1,000 μM and with resin at 2.0, 1.0, 0.5, 0.25 and 0.0 g (weight of resin per 100 mL of medium) of Amberlite XAD-16 resin (Rohm and Haas, Philadelphia, PA, USA) at 28 °C with shaking (150 r.p.m.) for 48 h (Khan et al., 2008; Gan et al., 2016). Amberlite XAD-16 is a non-ionic, hydrophobic, cross-linked polyaromatic resin. The media supernatants were separated from the resin by centrifugation and by transferring the supernatants to a new extraction flask. Supernatants and resin were extracted with acidified ethyl acetate (aEtOAc) 1:1 v/v (1 mL of glacial acetic acid per 200 mL of ethyl acetate) for 60 min with shaking (150 r.p.m.). The extracts were then centrifuged to separate the aqueous or resin and ethyl acetate phases. The ethyl acetate phase was recovered (1x extract) and dried in a Savant Speed Vac and one- and twenty-fold concentrated extracts were prepared and used in AHL detection bioassays and for LC-MS/MS analysis.

Biosensor detection and one- and two-dimensional thin layer chromatography for AHL visualization

Reverse-phase (RP) one-dimensional (1-D) TLC plates were used to determine AHL signal profiles. Concentrated acidified ethyl acetate (aEtOAc) extracts were spotted on to the C18 RP-TLC plate (EMD Chemicals Inc., Gibbstown, NJ, USA) origin in 2-mL volumes and representing from 0.5 to 2-mL supernatant equivalents. Plates were developed in a 70:30 (v/v) methanol: water mobile phase, dried and AHLs were detected as previously described (Scott et al., 2006; Gan et al., 2016). Bioluminescence produced by the A. tumefaciens A136 traR, PtraI::luxCDABE-based biosensor strain (Bernier, Beeston & Sokol, 2008) overlaid on the chromatograms was detected with a Bio-Rad charge coupled device (ccd) ChemiDoc MP system at different sensitivity settings to detect AHL signals with appropriate reference compounds. This involves determining and comparing retardation factors (Rf) of unknown samples with AHL reference compounds (Shaw et al., 1997).

RP two-dimensional (2-D) TLC for AHL separation and detection was developed in our laboratory (Gan et al., 2016) and performed by initially spotted one sample onto the bottom left corner of the C18 RP-TLC plate. The amount needed was estimated based on the AHL signal strength obtained from independent 1-D RP-TLC runs. The spotted TLC plate was eluted with 70:30 (v/v) methanol: water as the first mobile-phase in a glass tank. The mobile-phase was allowed to rise to the top of the TLC plate before removing the plate to dry overnight. Then, the TLC plate was rotated 90° counterclockwise, placed into a tank with 25:75 (v/v) 2-propanol: water as the second mobile-phase until it reached the top of the TLC plate as previous described by our laboratory (Gan et al., 2016). After drying, the TLC plate was overlaid with TraR-dependent Agrobacterium biosensor strain A136 using the process as used for 1-D TLCs (Scott et al., 2006; Gan et al., 2009, 2016).

AHL identification and quantification by liquid chromatography-electrospray ionization-tandem mass spectrometry (LC-ESI-MS/MS)

The Shimadzu series 10AD VP LC system was used to carry-out chromatography. A Phenomenex Gemini C18 HPLC column (3.0 μm, 100 × 3.0 mm) with an appropriate guard column was used and the column oven was set at 50 °C. Phase A (mobile) was 0.1% (v/v) formic acid in water, and phase B (mobile) consisted of 0.1% (v/v) formic acid in methanol. The chromatographic separation was carried-out at a flow rate of 450 μL/min, with a gradient initially at 10% B and increased linearly to 99% B over 12 min and remained at 99% B for 1 min. A rapid decrease to 10% B occurred over 0.1 min, and stayed at this composition for 1.9 min. Total run time per sample was 15 min. All methods are as previously described (Gan et al., 2016).

An Applied Biosystems Qtrap 4000 hybrid triple-quadrupole linear ion trap mass spectrometer was used with an electrospray ionization (ESI) interface. Analyst software was used to control the instrument, and for data collection and analysis. The following parameters were set and included: auxiliary gas: 15.0, curtain gas: 20.0, ion source potential: 5,000 V, nebulizer gas: 20.0 and temperature: 450 °C (Gan et al., 2016).

Sample preparation and AHL standards

Synthetic standards of C4, C6, C8, C10, C12, C14, 3-oxo-C4, 3-oxo-C6, 3-oxo-C8, 3-oxo-C10, 3-oxo-C12, 3-oxo-C14, 3-OH-C4, 3-OH-C6, 3-OH-C8, 3-OH-C10, 3-OH-C12 and 3-OH-C14 AHLs were synthesized according to established procedures (Chhabra et al., 1993; Gan et al., 2016). Dried extracts were stored at −20 °C. Prior to analysis, each sample extract was reconstituted in 100 μl of methanol +0.1% (v/v) formic acid. The injection volume was 5 μl (Gan et al., 2016).

Analysis method

Initial analysis was conducted with the MS operating in precursor ion scan mode screening for precursor ions that give rise to a product ion of m/z = 102 (a fragment ion that is common to all AHLs), upon collision induced fragmentation (Table S2). Comparison of detected peak areas with an AHL mix sample of known concentration was used to gauge a useful calibration range for the subsequent quantification of detected AHLs. Samples were re-run with the MS in MRM (multiple reaction monitoring) mode, analyzing the LC eluent for specific AHLs detected in the previous analysis. The quantification was conducted by comparing peak areas of detected peaks with a six-point calibration curve constructed by analyzing (in triplicate) mixed AHL calibration samples containing C8, 3-OH-C8 and 3-OH-C10 AHLs at 0.5, 1.0, 2.0, 5.0, 10 and 20 μM (Gan et al., 2016).

Results

Whole genome sequencing of N. sp. Rr2-17 identified a single luxI/ luxR pair homologs (novI, novR)

The draft genome of strain Novosphingobium. sp. Rr2-17 (Gan et al., 2012) consists of 4,539,029 bps (148 × coverage), has a GC content of 62.7% and consists of 166 contigs (N50 of 130 kb with largest being 318 kb) (Table S3). The nucleotide sequence accession number is GenBank: AKFJ00000000.1 (Gan et al., 2012). Anti-SMASH analysis (Blin et al., 2019) of N. sp. Rr2-17 identified a single luxI homolog and its luxR cognate receptor was found canonical luxI luxR pair (novI, novR) genetically linked to each other (Table S4). By performing BLAST searches against the curated LuxI homologs, a putative AHL synthase was identified in strain Rr2-17, locus tag WSK_3264. We propose the name novIspR1 for locus tag WSK_3264 of N. sp. Rr2-17.

NovIspR1 alignment with LuxI homologs of the Novosphingobium genus and phylogenetic tree analysis

Alignment of the NovIspR1 protein sequence with known LuxI homologs showed that all the homologs, including the luxI homolog of Rr2-17 strain, NovIspR1, contain the highly conserved amino acid signatures which are required for the function of AHL synthases (Fig. S1, asterisk marks) and includes three highly conserved amino acids, Arg24, Phe28 and Trp34. Among some autoinducer proteins within the genus previously identified, NovIspR1 share identity with three LuxI homologs of N. subterraneums DSM 12447, NovINsub1, NovINsub2 and NovINsub3 at 39.6%, 49.5% and 50.3% for locus tags NJ75_2841, NJ75_2498 and NJ75_4146, respectively (Gan et al., 2016).

Phylogenetic analysis of publicly available Novosphingobium LuxI homologs, including NovIspR1, revealed that NovIspR1 forms a sister group with the LuxI homologs of N. pentaromativorans and N. sp. TCA1 (Uniprot accession codes A0A2W5NAM5 and A0A6H9HEL6, respectively), with strong support (Fig. 2). While certain Novosphingobium species possess multiple copies of LuxI homologs, our findings indicate that these homologs are often distantly related, occupying different phylogenetic clades. An intriguing observation is the putative LuxI homolog of N. resinovorum (A0A1D8AGG5), which, despite sharing the signature conserved amino acids of an acyl-homoserine lactone synthase (Fig. 2), appears basal to the other Novosphingobium LuxI homologs, showing a relatively long branch length. This highlights a potential unique evolutionary trajectory for homolog novIspR1.

Figure 2 In-silico validation of the NovIspR1 based on maximum likelihood tree inference.

Evolutionary relationships among putative LuxI homologs from the genus Novosphingobium. The maximum likelihood tree was rooted with LuxI homologs from R. radiobacter and A. fabrum as the outgroups. Node labels indicate SH-like branch support values and branch lengths represent number of substitutions per site.

NovIspR1 protein of epiphytic crown gall strain N. sp. Rr2-17 produces multiple AHL signals

Culture extracts prepared from wild-type N. sp. Rr2-17 activated two of the four AHL-dependent whole cell bacterial biosensors tested (Table 2). 1-D RP-TLC separation of the culture extract followed by AHL detection using the Agrobacterium-based TraR-based bioluminescence biosensor led to the detection of four putative AHL signals (Fig. 3, lane 4).

Table 2 Production of N-acyl-homoserine lactones by Novosphingobium sp. Rr2-17 strain as assayed by four different AHL-dependent biosensor strains1.

Strain	AHL-dependent biosensor strain	
	AhyR1	LuxR	TraR	LasR	
N. sp. Rr2-17	- 2	+	+++	–	
Notes:

1 Abbreviations include: AhyR, AHL receptor from Aeromonas hydrophilia; LuxR, from Vibrio fisheri; TraR, from Agrobacterium tumefaciens; LasR, from Pseudomonas aeruginosa.

2 Scores for biosensor detection of AHL in N. sp. Rr2-17 strain extracts are based on the following criteria: −, < 2-fold higher than background levels of relative light units (RLU) bioluminescence; +>2-fold higher than background RLUs; ++ >50 to 75-fold higher than background RLUs; +++ >75-fold higher than background in RLUs.

Figure 3 Acyl homoserine lactones (AHL) produced by wild type strain Rr2-17 in chromatograph of extract and AHL standards and detection with Agrobacterium biosensor strain NTL4(pZLR4) overlay.

AHL standards include: substituted AHL signals: 3-hydroxy-C6-AHL (OH-C6, lane 1), 3-hydroxy-C8-AHL (OH-C8, lane 3) and unsubstituted AHL signals C6-AHL (C6, lane 1) and C8-AHL (C8, lane 3) as previously described (Gan et al., 2009). Ethyl acetate extract (EtOAc) of culture supernatants of strain Rr2-17 (1.0 ml of 2.5 × EtOAC extract). E.coli JM109 (pSRKKm::novISpR1 liquid cultures were grown overnight in the presence of 0, 10, 100 or 1,000 μM of IPTG inducer and culture supernatants were extracted with EtOAc. Lanes 5 to 8 each represent 1 μl of 1x extracted overnight broth cultures. Each dot represent the center of signal. The sample loading origin, (O) is indicated at bottom of chromatogram.

Further, dose-dependent accumulation of AHLs was observed from growth culture supernatants of JM109 (pSRKKm::novIspR1) at different IPTG-inducer concentrations. Culture supernatants from JM109 (pSRKKm::(novIspR1) showed an increase in AHLs from background levels (no inducer IPTG) with as little as 10 μM of IPTG and continued to increase when exposed to 100 and 1,000 μM of IPTG inducer (Fig. 3, lanes 5–8). As expected, the highest accumulation of AHLs was observed at 1,000 μM of inducer IPTG and this level of inducer was used in all further experiments.

Inducible over-expression of novIspR1 in wild type parental strain and accumulation of AHLs using binding resin Amberlite XAD-16

Inducible over-expression of novIspR1 in the wild-type parental strain which contains the native novIspR1 was expected to enhance the biosynthesis and allow detection of additional AHLs signals. To test this, plasmid pSRKKm::novIspR1 was introduced into the parental strain. Overexpression of an additional copy of the novIspR1 gene combined with varying amounts of XAD-16 resin in the growth cultures influenced AHL accumulation and recovery (Fig. 4). IPTG induction consistently increased AHL levels in a concentration-dependent manner, and higher resin concentration enabled recovery of more AHL signals (Figs. 3 and 4). An example of the AHL signals separated by 1-D TLC shows a significant increase in the intensity and diversity of AHL signals produced, extracted and visualized from homologous over expression of novIspR1 (Fig. 4B). Saturating levels of resin occurred at 1% as determined by similar detection intensities of AHL signals recovered and visualized by TLC biosensor overlay from extracted broth (Fig. 4A) and extracted resin (Fig. 4B). This may indicate that the AHL-dependent biosensor is near or at a saturated response, and that any additional signal has little effect on the response / output, when using the lacZ as the reporter gene.

Figure 4 Acyl homoerine lactone (AHL) profiles after TLC-based chromatography and AHL detection with biosensor NTL4(pZLR4).

Extracts were recovered from broth cultures of Rr2-17 (pSRKKm::novI-Sp-R1) and Rr2-17 containing empty control plasmid pSRKKm (pSRKKm only). Extracts were recovered from broth cultures of Rr2-17 (pSRKKm::novISpR1) and Rr2-17 containing empty control plasmid pSRKKm (pSRKKm only) containing increasing concentrations of binding resin XAD-16. Extracts of the broth culture media without binding resin (A) and extracts of the binding resin after separated from the broth culture (B). Binding resin XAD-16 was added to broth culture media at 0, 0.5, 1.0 and 2.0%, (white triangles), inoculated and grown overnight. Standards on TLC chromatograph included: unsubstituted AHLs: C6, C8, C10, and C12 (S1), and of hydroxy substituted OH-C6 and OH-C8 (S2) as previously described (Gan et al., 2009).

To validate these findings, a second Agrobacterium-based biosensor, A136 (pCF218) (pMV26), that also uses the identical AHL receptor, TraR, but provides an output response signal of bioluminescence was employed. Dis-diffusion experiment confirmed that increasing resin concentrations enhanced recovery of total AHLs. (Fig. 2A). Quantification of the total AHLs show a decrease in the amount of AHL signals remaining in the culture media, indicating that as they were captured by the resin (Figs. S2B, S2C).

LC-MS/MS, 2D-TLCs and XAD-16 resin to determine AHL signals

A complement of nine AHL signals were found to be produced by the wild type strain and five were identified by over expression in E. coli JM109 strain (Table 3). The LC-MS/MS output traces from this analysis are shown (Fig. S3) and retention times expected and observed are shown in Table 3. LC-MS/MS analysis revealed that the most abundant AHL produced by novIspR1 was 3-OH-C8, followed by 3-OH-C10 and C8. The ratio of 3-OH-C8 to 3-OH-C10 differed by at least 5-fold in control cultures (no resin) and varied between 5.8 and 11.5 in resin extracts, depending on resin concentration (Fig. 5, Table S5). Increasing the resin concentration from 0.25% to 2.0% reduced the 3-OH-C8 to 3-OH-C10 ratio, with a decrease to 5.8 and 8.0 at 2.0% resin for Rr2-17 (pSRKKm::novIspR1) and Rr2-17 (pSRKKm), respectively. Overexpression of novIspR1 in wild-type Rr2-17 combined with 2.0% resin increased AHL recovery by 7- to 9.3-fold for C8, 3-OH-C8, and 3-OH-C10 compared to the control strain (Table S5).

Table 3 Acyl-homoserine lactone species, acyl chain lengths and observed retention time of the characteristic total molecular ions of m/z for wild type Rr2-17 strain and DH5α (pSRKkm::novISpR1).

	
Strain or luxI homolog	Number of AHLs detected	R1
Substitution at the 3rd carbon	R2
Acyl chain length (total number of carbons)	Retention time/min of signal standard	Retention time/min of signal observed	
Rr2-17	9	OH	C6	3.93	3.97	
		Unsubstituted, OH	C8	5.20
4.49	5.20
4.49	
		Unsubstituted, OH	C10	5.87
5.05	5.83
5.05	
		Unsubstituted, OH	C12	6.65
5.72	6.65
5.72	
		Unsubstituted, OH	C14	7.44	7.44	
				6.50	6.50	
DH5a
(pSRKkm::NovIspR1)	5	Unsubstituted, OH	C8	5.20
4.49	5.16
4.49	
		OH	C10	5.05	5.05	
		OH	C12	5.72	5.72	
		OH	C14	6.50	6.50	

Figure 5 Concentration of the three most abundant AHL signals produced from NovISpR1.

Overexpression of novISpR1 in parental strain Rr2-17 and recovery of the acyl homoserine lactones (AHLs) sequestered in resin during culture growth culture. Concentration of the three most abundant AHL signals, OH-C8, OH-C10 and C8, are shown after recovery from resin and culture media extracts prepared from cultures of Rr2-17 (pSRKKm::novISpR1 containing XAD-16 resin at 0.25%, 0.5%, 1.0%, and 2.0%. Controls include: culture extracts from growth of Rr2-17 (pSRKKm::novISpR1) and Rr2-17(pSRKKm) in the absence of resin.

2-D RP-TLC for enhanced signal detection

The application of 2-D RP-TLC facilitated the separation and detection of multiple AHLs from Rr2-17 (pSRKKm::novIspR1). A total of eleven AHL spots were identified, with seven clearly visible and four faint signals detected in the novIspR1 resin extracts (Fig. 6A, Fig. S3). In contrast, only one faint signal was found in the culture extracts of Rr2-17 (pSRKKm) (Fig. 6B), with no additional spots in its resin extracts, highlighting the impact of novIspR1 overexpression.

Figure 6 Two-dimensional thin layer chromatography (2-D TLC) of Rr2-17 (pSRKKm::novIspR1) resin extracts (A), and unsubstituted (B) and OH substituted.

2-D TLCs were dried and overlayed with a medium-agar culture of the TraR-dependent biosensor A136, incubated for 36 h and imaged by a charge coupled device (Bio-Rad). 2-D TLC of the Rr2-17 (pSRKKm::novI) extract after imaging with the A136 biosensor (A) and the same with black and white detection (B) and at the same sensitivity used for the color detection shown in panel A. 2-D TLC of C6, C8, C10, and C12 as unsubstituted standards (C). 2-D TLC of 3-OH-C6, 3-OH-C8, 3-OH-C10, and 3-OH-C12 as OH substituted standards (D). The green circle in each panel represents the origin of the resin extract (A, B) and origin of pure standards in 2-D TLC bioassay (C, D).

Saturation at 1% resin was indicated by similar detection intensities of AHL signals from both extracted broth (Fig. 6C) and resin (Fig. 6D). The varying visibility of signal spots in resin extracts from Rr2-17 (pSRKKm::novIspR1) compared to those from Rr2-17 (pSRKKm) is further illustrated in Fig. S4. LC-MS/MS analyses confirmed that overexpression of novIspR1 in the wild-type strain produced at least nine AHLs, including C8-C10, C12, C14, and five hydroxyl-substituted AHLs (3-OH-C6, -C8, -C10, -C12, -C14). Additionally, two potentially uncharacterized signals activated the TraR receptor in whole cell biosensor overlays, indicating a diverse array of AHLs greater than typically observed in bacteria with a single luxI homolog.

Discussion

In previous studies from our lab, a Tn5 insertional mutant of the plant epiphytic strain Novosphingobium sp. Rr2-17 (named strain Hx 699) showed ten-fold reduction in AHL signal production (Gan et al., 2009). The disrupted gene was identified in a RelA/SpoT homologue called a rsh gene, a gene involved in the stringent response (Gan et al., 2009). Strain Hx 699 showed a hypomucoid phenotype and promoted cell aggregation (Gan et al., 2009). In addition, we have determined the sequence of the whole genome of Rr2-17 (Gan et al., 2012) and confirmed the presence of a single luxI/R pair homolog (named novI/R) and its genome neighborhood (Gan et al., 2013). In this work, the overproduction of AHLs by overexpression of second copy of the novIspR1 in the parental stain Rr2-17 showed the same phenotype in comparison to wild-type Rr2-17 strain and provide a functional validation of the single novI homolog, novIspR1. Several lines of evidence are provided including the cloning and expression of the novIspR1 in the wild-type parental strain, the use of an inducible promoter for novIspR1 overexpression and the use of Amberlite XAD-16 resin to sequester AHL signals in actively growing and induced bacterial cultures. The number of AHL signal produced and their chemical identities were characterized by multiple AHL-dependent signal detection bioassays using 1-D and 2-D RP-TLC and LC/MS-MS analyses.

The characterization of the AHL signals extracted from resin and residual broth media after induction followed by detection using two TraR-based AHL-dependent biosensors, one with lacZ reporter and the second with the luxCDABE reporter coupled to detection by 1-D and 2-D RP-TLC enabled two additional induced signal spots to be distinguishable only in the resin extracts from the wild type parental strain Rr2-17 that over expresses the novIspR1 in the inducible system (Fig. 6A compared to Figs. 6B–6D). Furthermore, resin and residual broth media extracts were characterized by LC-EI MS/MS (confirmed authenticity of nine novIspR1-produced AHL signals.

Although in-silico analyses have offered valuable insights into NovIspR1 as an AHL synthase, predicting the specific number and types of AHLs solely based on its amino acid sequence remains elusive. We address this gap by providing crucial functional validation, complementing existing in-silico analyses, and potentially enabling accurate predictions of AHL synthesis capacity for other LuxI homologs that are closely related to NovIspR1. Regrettably, despite the growing availability of publicly accessible genomes and complete open reading frames of numerous predicted luxI homologs in Novosphingobium, functional validation has only been accomplished for LuxI homologs from two Novosphingobium strains, to date. Consequently, the identification of critical amino acid residues associated with the substrate range of AHL synthases among Novosphingobium strains remains limited.

The whole-cell Agrobacterium-based TraR-dependent receptor biosensor is known to respond to many AHLs of different length and substitutions of the acyl side chain and has been referred to as a broad-range AHL biosensor (Cha et al., 1998). This broad-range feature is present in the two TraR-receptor AHL-dependent biosensors used in this work with the reporter output dependent upon the specificity of the TraR receptor protein (Steindler & Venturi, 2007). In addition, a direct detection method such as LC coupled with MS/MS, as shown here, is necessary to unambiguously validate the production of AHLs by the N. sp. Rr2-17 strain. Here both LC-MS/MS analyses and 1-D and 2-D TLC bioassays revealed that expression of novIspR1 produces a suite of nine AHLs, four unsubstituted: C8-, C10-, C12-, C14-HSL as well as five AHLs with OH substitutions at the third carbon and includes 3-OH-C6, 3-OH-C8, 3-OH-C10, 3-OH-C12 and 3-OH-C14. This constitutes many AHL signals produced from a single luxI homolog and as a result this strain of Novosphingobium may be of particular interest.

The LC-MS/MS data suggests that the major AHL signal being produced is OH-C8. OH-C8 HSL was found to be produced at a concentrations 10 times higher than the next most abundant signal, OH-C10 HSL (Table S5) and is consistent with our previous work with three homologs of N. subterraneum DSM 12447 (Gan et al., 2016). It was also found that the overexpression of a second copy of novIspR1 in the wild type Rr2-17 strain increased the production of each of the measured signals several times with OH-C8 being found at five times the concentration of OH-C10. These increases are also shown qualitatively in the increases in light production by the AHL-dependent biosensor A136 (CF218) (pMV26) as shown in Figs. 2 and 3. While the other AHL signals detected could not be quantified due to lack of synthetic standards for dose response curves, there was still an increase in intensity of those signals on the TLCs (Fig. 2, Fig. S2). The 2-D TLCs also readily show the nine AHLs which is consistent with the number of AHLs detected using LC-MS/MS analysis.

By increasing volumes of XAD-16 resin in the actively growing cultures useful information for future extractions has been validated. There is a noticeable level of increased AHL concentration inextracts from resin, separated by TLC and AHL detection by whole-cell biosensors but it diminishes at certain resin concentrations. We believe this may be due to the biosensor being at saturated or saturated response levels, so additional signal has little effect on the reporter genes. There is a significant decrease in the amount of signal left in the broth media extract indicating additional AHL signal could be sequestered (bound by) with increasing amounts of resin. Overall, it does not appear that it is necessary to use larger volumes of resin to enhance the isolation of AHL signals (Neumann et al., 2013). In similar studies with Dinoroseobacter shibae and Photorhabdus luminescens, the use of 2.0 weight percent of resin in the liquid culture was determined the optimal concentration for signal yield (Neumann et al., 2013; Brachmann et al., 2013). Another related study with a marine mesorhizobium, showed that the use of the adsorber resin Diaion HP21enhanced the recovery and characterization of novel long-chain AHLs (Krick et al., 2007; Wagner-Döbler et al., 2005).

In this work, the LC-MS/MS analysis supports the fact that there is far more signal present in the resin extract with most signal bound to the resin as evidenced by the low residual of AHLs detected and identified in the broth extract. This also will enable the use of less ethyl acetate as the extraction solvent, in the preparation of AHL samples for analysis. This is consistent with the use of smaller sample extraction volumes due to the use of AHL binding resin to enable a more sustainable approach in the research laboratory.

Conclusions

The grapevine crown gall tumor strain Rr2–17 contains a single AHL synthase homolog that produces multiple AHLs of various chain lengths of either unsubstituted or containing an OH substitution at the 3rd carbon. Why novIspR1 can produce multiple AHLs and how this influences the biology of Rr2-17 on grapevine crown gall tumors is largely unknown. The following insights will enable a framework for additional work with this bacterium. First, this AHL synthase may contain a rather broad specificity for its acyl carrier counterpart as previously reported for some AHL synthases (Gao et al., 2005; Steindler et al., 2009; Montebello et al., 2014; Jin et al., 2021). Second, other AHL synthase encoding genes were not identified due to sequence divergence. Third, since our genome sequence is a draft, it is possible that one or more of the AHL synthase genes were lost in sequence gaps. Additional research entailing the identification and the modification of structural features of novIspR1 including specific residues involved in acyl-ACP substrate preference encoded by novIspR1 in strain Rr2-17 and assessment of AHL production in the mutant strains will be important to validate the AHL signal profile of Rr2-17. Correspondingly, the construction of a novIspR1 mutant strain of Rr2-17 deficient in AHL production will be important for future work involving whole cell transcriptome sequencing to directly provide insights into the quorum sensing regulatory cascade in Novosphingobium strain Rr2-17 and genomic features of bacterial adaptation to Agrobacterium vitis crown gall tumors of vineyards.

Supplemental Information

Supplemental Information 1 In-silico validation of the NovISpR1 based on amino acid alignment.

Alignment of NovISpR1 (green arrow) and other putative Novosphingobium LuxI homologs with the functionally confirmed LuxI homologs from Rhizobium radiobacter and Agrobacterium fabrum (lines 1 and 2, above green arrow). Highly conserved amino acid residues were highlighted.

Supplemental Information 2 Bioluminescent quantification of total AHLs from disc-diffusion bioassay using bioluminescent biosensor A136.

(A) Image was captured using Bio-Rad ChemiDoc Mp Imaging System with an exposure time of 35 seconds. Bioluminescent responses are shown of total AHLs with increasing concentrations of XAD resin (XAD) extracts and broth (BE) extracts from broth media cultures of Rr2-17 (pSRKKm) under IPTG inducible conditions and the same from broth media cultures of Rr2-17 (pSRKKm::novISp-R1) under IPTG inducible conditions. (B) Bioluminescent quantification of total AHLs extracted from resin and culture from broth cultures containing different concentrations of resin (2.0. 1.0, 0.5, 0.25 %) of Rr2-17 (pSRKKm) under IPTG inducible conditions and (C) the same of cultures of Rr2-17 (pSRKKm::novIspR1). Abbreviations: Broth extracted, BE; Resin XAD-16 extracted, XAD.

Supplemental Information 3 LC-ESI-MS/MS chromatograms of nine AHLs produced by Rr2-17 strain.

Supplemental Information 4 Overexpression of the novIspR1 in wild type strain Rr2-17, two-dimensional (2-D) TLC and bioluminescence detection in color and black and white (B&W) for AHL signal spot detection.

2-D TLC of the Rr2-17(pSRKKm::novIspR1) resin extract (A, B) and culture extract (C, D) after imaging with the bioluminescent biosensor A136 in color (A, C) and black & white (B&W) detection (B, D). 2-D TLC of the Rr2-17 (pSRKKm) resin extract (E, F) and culture extract (G, H) after imaging with the bioluminescent biosensor A136 in color (E, G) and B&W detection (F, H). Blue arrows indicate four and two signal spots observed in panels A, B and C, D, respectively, that are not visible in resin or culture extracts from Rr2-17 (pSRKKm) (E, F and G, H), respectively. Origin of extract is as noted in figure 6.

Supplemental Information 5 Whole cell biosensor strains used in this work.

1 C4-HSL, N-butanoyl-homoserine lactone; C6-HSL, N-hexanoyl-homoserine lactone; 3-oxo-C6- HSL, N-3-oxo-hexanoyl-homoserine lactone; 3-oxo-C8-HSL, N-3-oxo-octanoyl-homoserine lactone; 3-oxo-C12-HSL, N-oxo-dodecanoyl-homoserine lactone.

Supplemental Information 6 Mass transitions used for the MRM detection of common AHLs.

Supplemental Information 7 Genome statistics of Novosphingobium sp. Rr2-17.

Supplemental Information 8 Secondary metabolite biosynthetic gene clusters identified by antiSMASH platform.

Supplemental Information 9 LC-MS/MS quantification of three acyl homoserine lactones in μM in resin (XAD) and residual broth culture (BE) and from broth culture without resin (BC).

This article is dedicated to Ernő Szegedi formerly of the Research Institute for Viticulture and Enology, Kecskemét, Hungary, and we thank him for his generosity over the years in sharing many bacterial isolates, including Novospingobium sp. Rr2-17.

Additional Information and Declarations

Competing Interests

Author Contributions

Data Availability

The authors declare that they have no competing interests. Han Ming Gan is employed by Patriot Biotech Sdn Bhd.

Han Ming Gan conceived and designed the experiments, performed the experiments, analyzed the data, prepared figures and/or tables, authored or reviewed drafts of the article, and approved the final draft.

Lucas Dailey performed the experiments, prepared figures and/or tables, authored or reviewed drafts of the article, and approved the final draft.

Peter Wengert performed the experiments, prepared figures and/or tables, authored or reviewed drafts of the article, and approved the final draft.

Nigel Halliday performed the experiments, authored or reviewed drafts of the article, and approved the final draft.

Paul Williams conceived and designed the experiments, analyzed the data, authored or reviewed drafts of the article, and approved the final draft.

André O. Hudson conceived and designed the experiments, analyzed the data, prepared figures and/or tables, authored or reviewed drafts of the article, and approved the final draft.

Michael A. Savka conceived and designed the experiments, performed the experiments, analyzed the data, prepared figures and/or tables, authored or reviewed drafts of the article, and approved the final draft.

The following information was supplied regarding data availability:

This Whole Genome Shotgun project for Novosphingobium sp. Rr2-17 is available at GenBank: AKFJ00000000.1; PRJNA158437.

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
