# Peer review of "Quorum sensing signals of the grapevine crown gall bacterium, Novosphingobium sp. Rr2-17: use of inducible expression and polymeric resin to sequester acyl-homoserine lactones"

_PeerJ, doi:10.7717/peerj.18657_

## Round 0.1 · original submission · Major Revisions

First of all, the authors must properly respond to all demands of the reviewers; and, secondly, I propose to work on compactness of the presentation. It seems that the paper is unnecessary long.

Reviewer 1 ·

Basic reporting

Manuscript by Han Ming Gan entitled ‘Quorum sensing signals in the grapevine…’


This manuscript describes the characterization of an AHL LuxI family synthase from the grapevine crown gall bacterium, Novosphingobium sp. Rr2-17. Characterization includes LC-MS of AHLs produced, overexpression of the luxI gene, novI, in E. coli and in the parent strain anad analysis using biosensors in TLC and in LC-MS. In addition, the binding to a resin was also assessed.


General comment:

The analysis of the AHLs produced by the luxI homolog of Novosphingobium sp. Rr2-17 has been performed using sensitive wide-range biosensors, resin and LC-MS. This LuxI produces many different AHLs at varying amounts. The characterization has been well performed. The study is rather limited in scope has only the AHL profiles have been described..


Specific comments:

1. Authors conclude that the most likely cognate (they use the word ‘native’) AHL as being C8-OH-AHL since it is the most abundantly produced. This is most likely the case however until response studies of LuxR cognate have been performed, this cannot be concluded. It is a pity that this simple experiment was not performed (most often done using the luxI gene promoter fused to a reporter gene in a plasmid) as it would have indicated whether the LuxR regulator is promiscuous due to the many AHLs produced by this strain.
2. I suggest that the genomic context of the luxI/R genes is presented, i.e. 5 kb on each side, and if it is conserved in Novoshingobium – very often the AHL QS system control the adjacent locus (loci).
3. No refence is made whether overexpressing the luxI gene in the parent strain results in any phenotype. A luxI mutant and its phenotypic study could have considerably added to this study.
4. The article could have been more concisely written, rather lengthy for the results presented.
5. Figures 2, 5, 6 and 7 can be moved to supplementary material.

Experimental design

The experiments presented have been well performed.

They are however rather limited in scope.

Validity of the findings

no comment.

already mentioned in my review in other sections.

·

Basic reporting

The authors have done good experimental work and have tried to give proper validation. However, there needs some improvement in the manuscript writing and experimental presentations. The English language also needs to be improved.

Experimental design

The methodologies used and experimental designs are relevant to the objectives of the experiments. However, corrections are required in the expression of units. For example, in lines 117 to 121, please check the units used i.e. antibiotic concentrations in solution. This needs to be checked throughout the manuscript. In Line 111, bacterial strains and plasmids used are listed in Table 1. It needs to be mentioned clearly from where they were procured.

Validity of the findings

Validation of the findings is done appropriately. However, Lines 461 to 463, 468-473 in Discussion section are not clear. Give relevant examples with references.

---

## Round 0.2 · accepted · Accept

We hope that the suggestions of the reviewers helped you to improve your manuscript. We also hope that you will be able fully explore these suggestions in your future research. It is pity that you are not able to check some issues right now.

Reviewer 1 ·

Basic reporting

The authors have addressed most of the comments raised by the reviewers. The conclusions drawn are now more in line with the results presented and the article is now more concise.

Experimental design

The experimental design remain limited but is clear and well performed.

Validity of the findings

The validity of the findings are clear; the mot relevant and important finding is the polymeric resin used to bind/sequester AHLs. This is now properly highlighted.